

# Prediction of cold chain loading environment for agricultural products based on K-medoids-LSTM-XGBoost ensemble model

Zhijie Luo[1,2,3,*], Wenjing Liu[1,*], Jianhao Wu[1], Huang Aiqing[1] and Jianjun Guo[1,2,3]

[1] Zhongkai University of Agriculture and Engineering, Guangzhou, China
[2] Smart Agriculture Engineering Technology Research Center, Zhongkai University of Agriculture and Engineering, Guangzhou, China
[3] Guangzhou Key Laboratory of Agricultural Product Quality and Safety Traceability Information Technology, Zhongkai University of Agriculture and Engineering, Guangzhou, China
* These authors contributed equally to this work.

## ABSTRACT

Cold chain loading is a crucial aspect in the process of cold chain transportation, aiming to enhance the quality, reduce energy consumption, and minimize costs associated with cold chain logistics. To achieve these objectives, this study proposes a prediction method based on the combined model of K-medoids-long short-term memory (LSTM) networks—eXtreme Gradient Boosting (XGBoost). This ensemble model accurately predicts the temperature for a specified future time period, providing an appropriate cold chain loading environment for goods. After obtaining temperature data pertaining to the cold chain loading environment, the K-medoids algorithm is initially employed to fuse the data, which is then fed into the constructed ensemble model. The model's mean absolute error (MAE) is determined to be 2.5343. The experimental results demonstrate that the K-medoids-LSTM-XGBoost combined prediction model outperforms individual models and similar ensemble models in accurately predicting the agricultural product's cold chain loading environment. This model offers improved monitoring and management capabilities for personnel involved in the cold chain loading process.

## INTRODUCTION

Cold chain logistics generally refers to a systematic engineering process that ensures the quality and minimizes losses of perishable products, including refrigerated and frozen foods, pharmaceuticals, and other temperature-sensitive items. This process encompasses every stage, from production, storage, transportation, and sales to consumption, wherein the products are consistently maintained in prescribed low-temperature environments. This system is designed to guarantee product quality and reduce losses, making it a vital component of the broader cold chain transportation field. With continuous advancements in science and technology, the emerging industry of cold chain logistics has experienced robust growth.

Corresponding author
Jianjun Guo, glgxbaobao@163.com

With the continuous improvement of income levels and the increasing consumption of the Chinese urban and rural residents, consumers have higher demands for food diversity, nutritional value, and taste. Additionally, the rapid rise of the fresh e-commerce market has propelled the cold chain logistics industry into a fast lane of development. However, with the widespread use of cold chain logistics, numerous issues have also been exposed. Due to the perishability and time sensitivity of fresh products, each link in the cold chain logistics is crucial for their preservation.

Within the cold chain loading process, among various environmental factors, the internal temperature of the vehicle compartment is a crucial factor influencing the quality of agricultural products. Sustaining the required temperature within the vehicle compartment continuously incurs significant resource consumption, leading to increased transportation costs. However, if the refrigeration system of the truck is only activated when fresh goods are ready to be loaded, there will be a time delay for the refrigeration system to reach the required temperature. During this period, there is an increased risk of fresh goods deteriorating, which subsequently raises transportation costs. Hence, it becomes a critical issue to determine the appropriate time to activate the refrigeration system of the truck, ensuring that the required temperature is reached before loading the goods, while also avoiding excessive energy consumption of the cold chain vehicle. This study investigates the temperature variations and corresponding time frames after activating the refrigeration system of the cold chain vehicle under different ambient temperatures. On the basis of accurate temperature prediction, assisting the cold chain sector in better controlling the timing of cold chain truck activation holds significant importance for ensuring the safety, quality, and cost reduction in the transportation of agricultural products. Extensive research has been conducted on cold chain transportation (*Gerber, Duarte & Alencar, 2020*; *Chen, Wu & Shao, 2020*; *Joshi et al., 2019*; *Liu et al., 2019*; *Lim et al., 2022*; *Zhao et al., 2010*; *Wang & Yan, 2019*; *Du, Shen & Zhang, 2020*; *Qi & Tai, 2021*; *Chen et al., 2022*; *Yuan et al., 2022*; *Xu et al., 2014*; *He & Yin, 2021*), yielding certain achievements. *Yuan et al. (2022)* proposed a method for multi-step prediction of temperature and humidity inside refrigerated compartments based on the combination of the K-means clustering (K-means) algorithm and the long short-term memory (LSTM) networks network. *Xu et al. (2014)* introduced an improved neural network model for comprehensive risk prediction in cold chain logistics. *He & Yin (2021)* proposed a mathematical computation approach utilizing neural network algorithms and grey prediction to explore the demands of cold chain logistics, aiming to optimize the automation and innovation of cold chain processes to adapt to emerging trends.

In recent years, deep learning methods in machine learning, with convolutional neural network (CNN) and recurrent neural network (RNN) as typical representatives, have achieved remarkable success in various prediction domains. LSTM networks, a special type of RNN, address the issues of vanishing and exploding gradients in RNN. This enables LSTM networks to learn long-term dependencies and improve the accuracy of predictions. *Wang et al. (2017)* utilized LSTM, a deep learning technique, to capture the spatiotemporal relationships among seismic events occurring at different locations for predictive purposes. *Yan et al. (2021)* incorporated past flow data, historical weather data, and weather forecast

data, utilizing an Attention-LSTM prediction model to forecast traffic flow values for the upcoming 12 h. *Li et al. (2021)* proposed an Encoder-Decoder Multi-step Trajectory Prediction technique (EDMTP) based on LSTM networks to address the trajectory and motion characteristics of weakly constrained non-cooperative targets. Another popular algorithm in prediction domains is eXtreme Gradient Boosting (XGBoost). *Wang et al. (2022a)* proposed a joint prediction model for photovoltaic power generation that incorporates time series and multiple features, exhibiting high prediction accuracy, strong generalization ability, and robustness against noisy data. *Zhong et al. (2019)* aimed to improve the accuracy of short-term traffic flow prediction for road sections by employing XGBoost as the core algorithm and selecting average travel time as the prediction indicator. *Avanijaa (2021)* utilized the XGBoost regression technique after transforming categorical attributes into the required format using the one-hot encoding method to predict housing prices.

The proposed combined prediction model in this experiment integrates the two aforementioned prediction models and conducts research and prediction on data in the cold chain loading environment, thereby improving the accuracy of the prediction results to a certain extent. *Tan et al. (2022)* proposed a combined model for short-term photovoltaic power prediction based on the XGBoost model and the LSTM network. This model integrates the two models using the reciprocal of errors method, resulting in improved prediction accuracy. *Wang et al. (2022b)* employed the reciprocal of errors method to construct a combined prediction model by weighting the prediction data from LSTM and XGBoost for short-term wind power prediction. *Zhang et al. (2023)* proposed a combined model, XGBoost-LSTNet, for road surface temperature prediction, leveraging the XGBoost model and the Long Short-Term Time Series Network (LSTNet) model. The proposed model effectively improves winter highway traffic safety and offers guidance for the management, maintenance, and prevention of accidents related to ice (*Zhang et al., 2023*). *Du, Yin & Li (2020)* proposed a feature-enhanced model, XGBoost-LSTM, for base station traffic prediction. Compared to similar algorithms, this combined model demonstrates superior performance. The LSTM-XGBoost combined models mentioned in the above articles demonstrate superior prediction accuracy compared to individual prediction models, highlighting the strong predictive performance achieved by the combination of XGBoost and LSTM models in various domains. As demonstrated earlier, LSTM and XGBoost models exhibit excellent characteristics in predicting the cold chain transportation environment. In this article, we organically combine these two prediction models, optimize their parameters using the particle swarm optimization (PSO) algorithm and exhaustive search, and propose a novel combined prediction model method (PAM-LSTM-XGBoost) for the cold chain loading field to accurately forecast the environmental temperature of the loading vehicle.

# SYSTEM FRAMEWORK AND HARDWARE DESIGN

This project presents the design of an Internet of Things (IoT)-based platform for predicting the environmental conditions in agricultural product cold chain transportation. The system schematic diagram is shown in Fig. 1. The platform is integrated into a cold

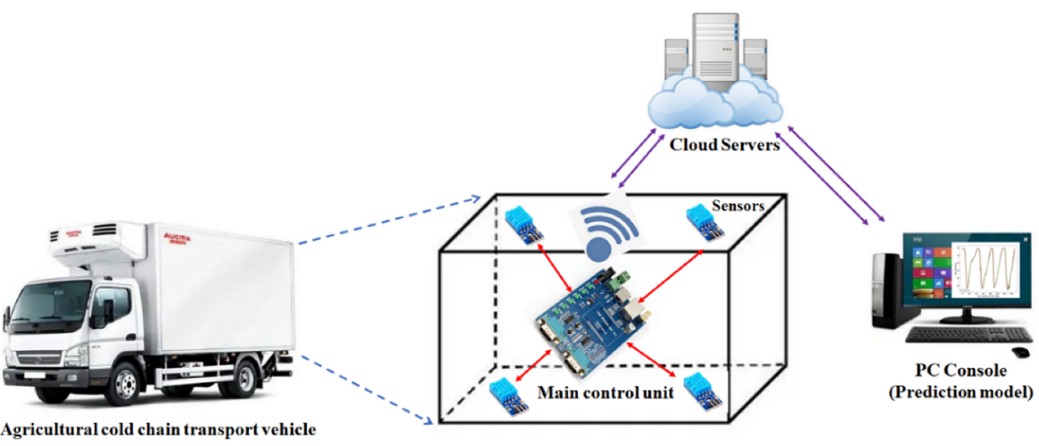

**Figure 1** Schematic of our platform.

chain transport vehicle, where the primary control unit utilizes the STM32F405 chip, a 32-bit processor manufactured by STMicroelectronics. To ensure stable operation for extended durations, the system incorporates four independent temperature and humidity sensors and is powered by an external portable power supply, providing over 100 h of continuous functionality. The refrigeration system of cold chain vehicles typically only contains temperature sensors at the air outlet or near the refrigeration unit, resulting in significant measurement inaccuracies and an inability to precisely capture the overall temperature within the cargo compartment. To monitor the temperature distribution throughout the compartment in real time and prevent potential temperature inconsistencies or anomalies, this study strategically places temperature and humidity sensors at four corners of the vehicle's cargo area. The sensors used in this study are DHT11 models, which incorporate temperature-sensing elements to measure the ambient temperature. For monitoring the internal environment of the cold chain vehicle, the DHT11 sensors are powered and their digital signals are subsequently decoded and processed to obtain accurate temperature data. Finally, the data collected from the four corners are aggregated using the K-medoids clustering algorithm to produce the temperature data required for prediction. The temperature and humidity data collected are stored in the FLASH memory of the chip and managed using a doubly linked list. The system transplanted the MQTT protocol to the main control unit, enabling the ESP8266 communication module to transmit this data to the ONENet cloud platform *via* the WiFi module every 3 min. A web-based control platform was developed to retrieve temperature and humidity data from the ONENet cloud. Additionally, the proposed prediction model handles this data. Finally, the prediction results and warning notifications are sent to a WeChat mini-program on mobile devices.

The system module diagram of the designed hardware system is presented in Fig. 2A, encompassing the main control unit, DHT11 temperature and humidity sensor, ESP8266 communication module, and portable power module. The main control unit incorporates the STMicroelectronics STM32F405 chip, a 32-bit processor. The DHT11 sensor interacts

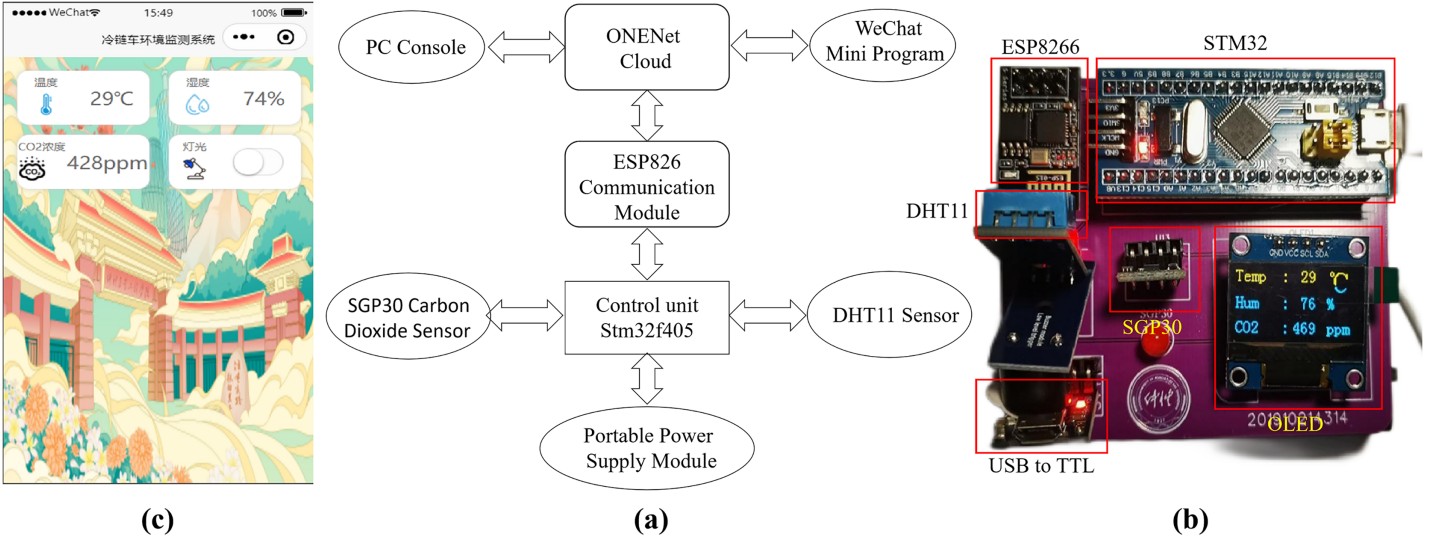

**Figure 2** (A) System module diagram of the designed hardware system; (B) Image of the designed hardware system; (C) Control interface of the developed WeChat mini-program on a mobile phone.

with the main control unit STM32 *via* a single bus connection. The control system utilizes the ES8266 module to facilitate data communication between the main control unit and the ONENet cloud platform. The prediction model is trained using temperature and humidity data obtained from the ONENet cloud platform and generates predictions accordingly. These prediction results are then uploaded back to the cloud platform. Additionally, a WeChat mini-program has been developed for mobile devices, providing users of the cold chain vehicle with access to the prediction results *via* the cloud platform. Users can make real-time adjustments to the cold chain vehicle's status based on the feedback received. To ensure stable operation within the refrigerated vehicle, the lithium battery power module supplies the required 3.3 V power voltage to the hardware system and its various modules.

## FORECASTING PRINCIPLES AND METHODS

### Data fusion using K-medoids algorithm

The K-medoids algorithm is an improvement upon the K-means algorithm. The K-means algorithm is susceptible to the influence of noise and outlier data, which can significantly distort the data distribution if extreme values or outliers are present. In contrast, the K-medoids algorithm employed in this experiment addresses this sensitivity issue by mitigating the impact of isolated points and noisy data on cluster assignment. It exhibits greater robustness compared to the K-means algorithm. The primary distinction between these two algorithms lies in their approach to determining cluster centroids. The K-means algorithm calculates the average value of the current cluster as the centroid, whereas the K-medoids algorithm selects a specific point within the cluster to serve as the centroid.

The steps involved in the K-medoids algorithm are as follows:

1) Randomly select k data points from the sample set as initial medoids.
2) Compute the distances from non-medoid data points in the sample set to the medoids.
3) Assign non-medoid data points to the cluster with the minimum distance to the medoid.
4) Randomly select a non-medoid data point $O_r$, calculate the sum of distances between this data point and all other data points D, if the calculated D is smaller than the original D, replace the original medoid with $O_r$.
5) Repeat steps (2), (3), and (4) until the difference between the calculated D of two adjacent iterations remains unchanged and each data point in the sample set has been traversed, indicating that the medoids no longer change; otherwise, continue with step 4.
6) Algorithm terminates, and the final medoids are outputted.

## Long short-term memory neural networks

Long short-term memory neural networks is a specialized type of RNN that possesses unique storage and forgetfulness functions, enabling it to address the drawback of limited short-term memory in traditional RNNs and learn long-term dependencies.

LSTM consists of three gates: the forget gate ($f_t$), the input gate ($i_t$), and the output gate ($O_t$). These gates control the cell state within LSTM by selectively incorporating or discarding information. The biases for the forget gate, input gate, and output gate are represented as $b_f$, $b_i$ and $b_o$, respectively. Furthermore, $b_C$ denotes the bias for the candidate vector. The weights associated with the forget gate, input gate, output gate, and candidate vector are denoted as $W_f$, $W_i$, $W_o$ and $W_C$, respectively. The activation functions used are the sigmoid function ($\sigma$) and the hyperbolic tangent function (tanh). $x_t$ represents the input at time step $t$, while $h_t$ and $h_{t-1}$ represent the outputs of the model at time steps $t$ and $t-1$, respectively. $C_t$ and $\tilde{C}_t$ correspond to the candidate vector at time step $t$ and the updated value of the candidate vector at time step $t$, respectively. Figure 3 illustrates the internal structure diagram of a temporal LSTM network.

(1) Forget gate

$$f_t = \sigma\left(W_f \cdot [h_{t-1}, x_t] + b_f\right) \tag{1}$$

(2) Input gate

$$i_t = \sigma(W_i \cdot [h_{t-1}, x_t] + b_i) \tag{2}$$
$$\tilde{C}_t = \tanh(W_i \cdot [h_{t-1}, x_t] + b_i) \tag{3}$$

(3) Unit

$$C_t = f_t * C_{t-1} + i_t * \tilde{C}_t \tag{4}$$

(4) Output gate

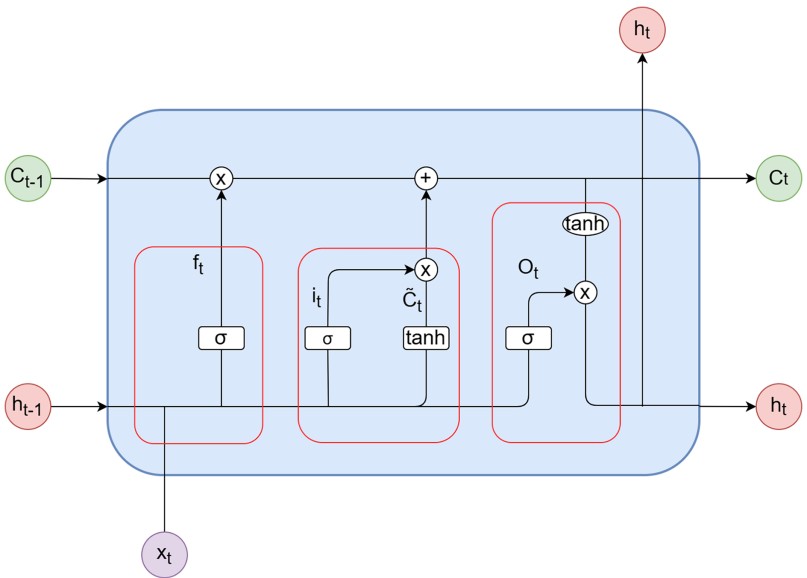

**Figure 3  LSTM unit.**           

$$O_t = \sigma(W_o \cdot [h_{t-1}, x_t] + b_o) \tag{5}$$
$$h_t = O_t \tanh(C_t) \tag{6}$$

## XGBoost algorithm

XGBoost, short for eXtreme Gradient Boosting, is an algorithm developed by Professor Tianqi Chen as an enhancement to the Gradient Boosting algorithm. Both XGBoost and gradient boosting decision trees (GBDT) utilize the concept of boosting, but their main difference lies in the definition of the objective function. While GBDT optimizes the objective function by expanding it to the first order using a Taylor series approximation, XGBoost expands the objective function to the second order while also incorporating additional regularization terms. This approach allows XGBoost to retain more relevant information from the objective function while controlling model complexity to prevent overfitting.

The XGBoost algorithm begins by generating a tree model at each iteration $t$. The tree model to be trained in the $t^{\text{th}}$ iteration is denoted as $f_t(x)$.

$$\hat{y}_i^{(t)} = \sum_{k=1}^{t} f_k(x_i) = \hat{y}_i^{(t-1)} + f_t(x_i) \tag{7}$$

$\hat{y}_i^{(t)}$ represents the predicted result of sample $i$ after $t$ iterations, while $\hat{y}_i^{(t-1)}$ represents the predicted result from the previous $t - 1$ trees. $f_t(x_i)$ denotes the tree model of the $t^{\text{th}}$ tree. The objective function consists of two components: the error function and the

regularization term. The objective function employed in this model and its associated regularization term are as follows:

$$OB_{j^{(t)}} = \sum_{i=1}^{n} l(y_i, \hat{y}_i^{(t)}) + \sum_{t=1}^{t} \Omega(f_t) = \sum_{i=1}^{n} l(y_i, \hat{y}_i^{(t-1)} + f_t(x_i)) + \Omega(f_t) + \sum_{t=1}^{t-1} \Omega(f_t) \qquad (8)$$

$$\Omega(f_t) = \gamma T + \frac{1}{2} \lambda \sum_{\Omega-1}^{T} w_o^2 \qquad (9)$$

$OB_{j^{(t)}}$ represents the objective function when constructing the $t^{\text{th}}$ tree. $l(y_i, \hat{y}_i^{(t)})$ denotes the loss function. $\hat{y}_i^{(t)}$ represents the predicted result of sample i after t iterations, while $\hat{y}_i^{(t-1)}$ represents the predicted result from the previous $t-1$ trees. $\Omega(f_t)$ represents the regularization term of the $t^{\text{th}}$ tree, where a smaller value indicates a lower complexity of the model, resulting in stronger generalization ability and more stable predictions.

The complexity of the tree model consists of two components: the number of leaf nodes T and the scores $w_o$ assigned to each leaf node. $\gamma$ and $\lambda$ serve as coefficients for the regularization term, controlling the number of leaf nodes and their scores.

The objective function is optimized using Taylor expansion, resulting in the following expression:

$$OB_{j^{(t)}} \approx \sum_{i=1}^{n} [l(y_i, \hat{y}_i^{(t)}) + g_i f_t(x_i) + \frac{1}{2} h_i f_t^2(x_i)] + \Omega(f_t) + constant \qquad (10)$$

Definition:

$$g_i = \partial_{\hat{y}^{(t-1)}} l(y_i, \hat{y}_i^{(t-1)}) \qquad (11)$$
$$h_i = \partial_{\hat{y}^{(t-1)}}^2 l(y_i, \hat{y}_i^{(t-1)}) \qquad (12)$$

The XGBoost algorithm utilizes a greedy algorithm to enumerate all possible tree structures and find the optimal structure that is subsequently added to the model. To prevent infinite tree generation, it is necessary to set stopping conditions. Common conditions include:

1) Setting a hyperparameter, max_depth, which stops tree generation when the tree reaches the maximum allowed depth to prevent overfitting.
2) Stopping the iteration and tree generation when the gain from a potential split falls below a specific threshold. At this point, the split can be disregarded. The threshold parameter is typically a coefficient of the number of leaf nodes T in the regularization term.

## Parameter optimization in particle swarm optimization

The PSO algorithm is a stochastic global optimization technique that utilizes interactions among particles to discover the optimal regions within a complex search space. The

algorithm draws inspiration from the migration and collective behavior observed in bird flocking during foraging. In the PSO algorithm, birds are abstracted as massless, dimensionless particles, and a group of particles represents a bird flock. The process of searching for the optimal solution within a defined area corresponds to the foraging behavior of the bird flock, achieved through systematic random movements of the particles. During the search for the optimal solution, particles collaborate and share information, with each particle following the vicinity of the current best particle to search for the optimal solution. In this process, particles continuously compare their own information with that of the best solution, enabling them to maintain an optimal state.

The PSO algorithm is described as follows:

In a D-dimensional region, there exist m particles forming a particle swarm $\{x_1, x_2, x_3, \ldots, x_m\}$. Each particle is characterized by its velocity $x_i = \{x_{i1}, x_{i2}, \ldots, x_{iD}\}$ and position $v_i = \{v_{i1}, v_{i2}, \ldots, v_{iD}\}$. The individual best position found by the $i^{\text{th}}$ particle is denoted as $P_i = \{P_{i1}, P_{i2}, P_{i3} \ldots, P_{iD}\}$, while the overall best position discovered by the entire particle swarm is represented as $P_{gBest} = \{P_{gBest1}, P_{gBest2}, P_{gBest3} \ldots, P_{gBestD}\}$.

During the $t^{\text{th}}$ iteration, each $i^{\text{th}}$ particle updates its velocity and position using the following formulas, as shown in Eqs. (13) and (14):

$$v_{id}^{t+1} = \omega v_{id}^t + c_1 r_1 \left( P_{id} - x_{id}^t \right) + c_2 r_2 \left( P_{gBestd} - x_{id}^t \right) \tag{13}$$

$$x_{id}^{t+1} = x_{id}^t + v_{id}^{t+1} \tag{14}$$

In the above equations, $i$ ranges from 1 to $m$, and $d$ ranges from 1 to $D$. The inertia weight $\omega$ represents the degree to which the particle retains memory of its previous velocity. A higher $\omega$ value enhances the particle's global search capability, while a lower $\omega$ value emphasizes its local search ability. The learning factors, $c_1$ and $c_2$, correspond to the particle's self-cognition ability and social sharing ability, respectively. These factors assist particles in converging towards local optimal points. The variables $r_1$ and $r_2$ denote random numbers. The algorithm terminates either when the maximum iteration count is reached or when the desired precision is achieved.

The flowchart of the PSO algorithm is illustrated in Fig. 4.

## A combined predictive model: K-medoids-LSTM-XGBoost

The ensemble prediction model integrates information from multiple individual prediction models and combines their results using specific methods. By extracting and integrating this information, the ensemble model leverages the strengths of different models, thereby effectively enhancing the accuracy of prediction results and bolstering the persuasiveness of experimental outcomes.

In this experiment, a weighted approach is employed to combine the two models, as demonstrated by the following formula:

$$f_t = w_1 f_{1t} + w_2 f_{2t}, t = 1, 2, 3, \ldots, n \tag{15}$$

Here, $f_{1t}$ denotes the prediction results derived from the LSTM model, while $f_{2t}$ represents the prediction results obtained from the XGBoost model.

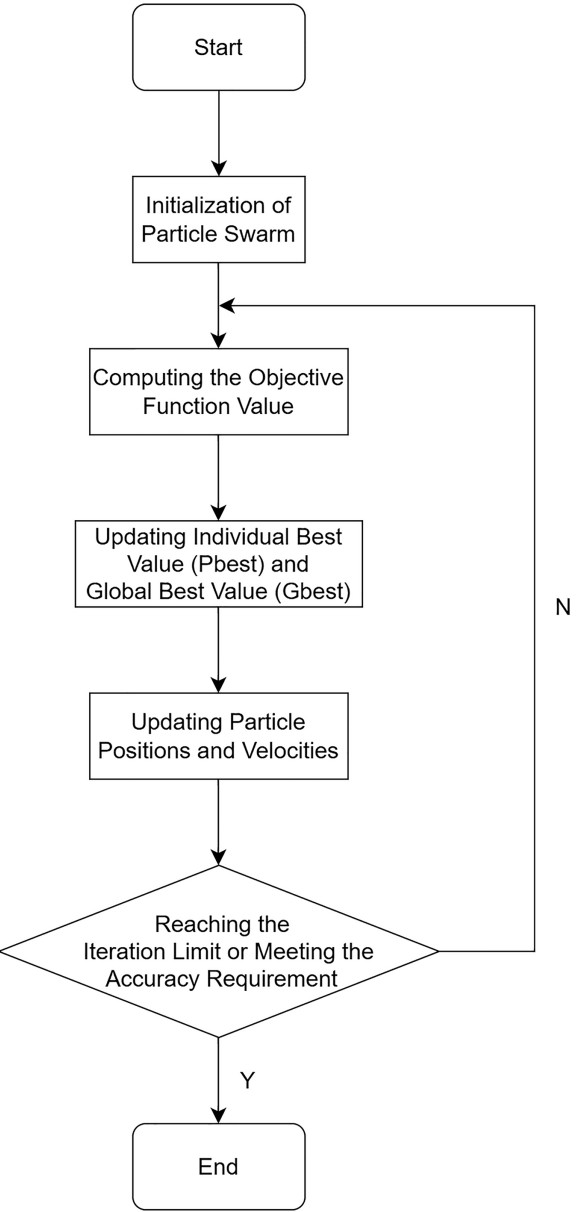

**Figure 4** **Flowchart of the PSO algorithm.**     

The weight coefficients, denoted as $w_i$, are determined according to the following equations:

$$w_1 = \begin{cases} \dfrac{\alpha_1}{\alpha_1 + \alpha_2}, \alpha_1 \geq \alpha_2 \\ \dfrac{\alpha_2}{\alpha_1 + \alpha_2}, \alpha_1 < \alpha_2 \end{cases} \tag{16}$$

$$w_2 = \begin{cases} \dfrac{\alpha_1}{\alpha_1 + \alpha_2}, \alpha_1 \geq \alpha_2 \\ \dfrac{\alpha_2}{\alpha_1 + \alpha_2}, \alpha_1 < \alpha_2 \end{cases} \tag{17}$$

**Peer**J Computer Science

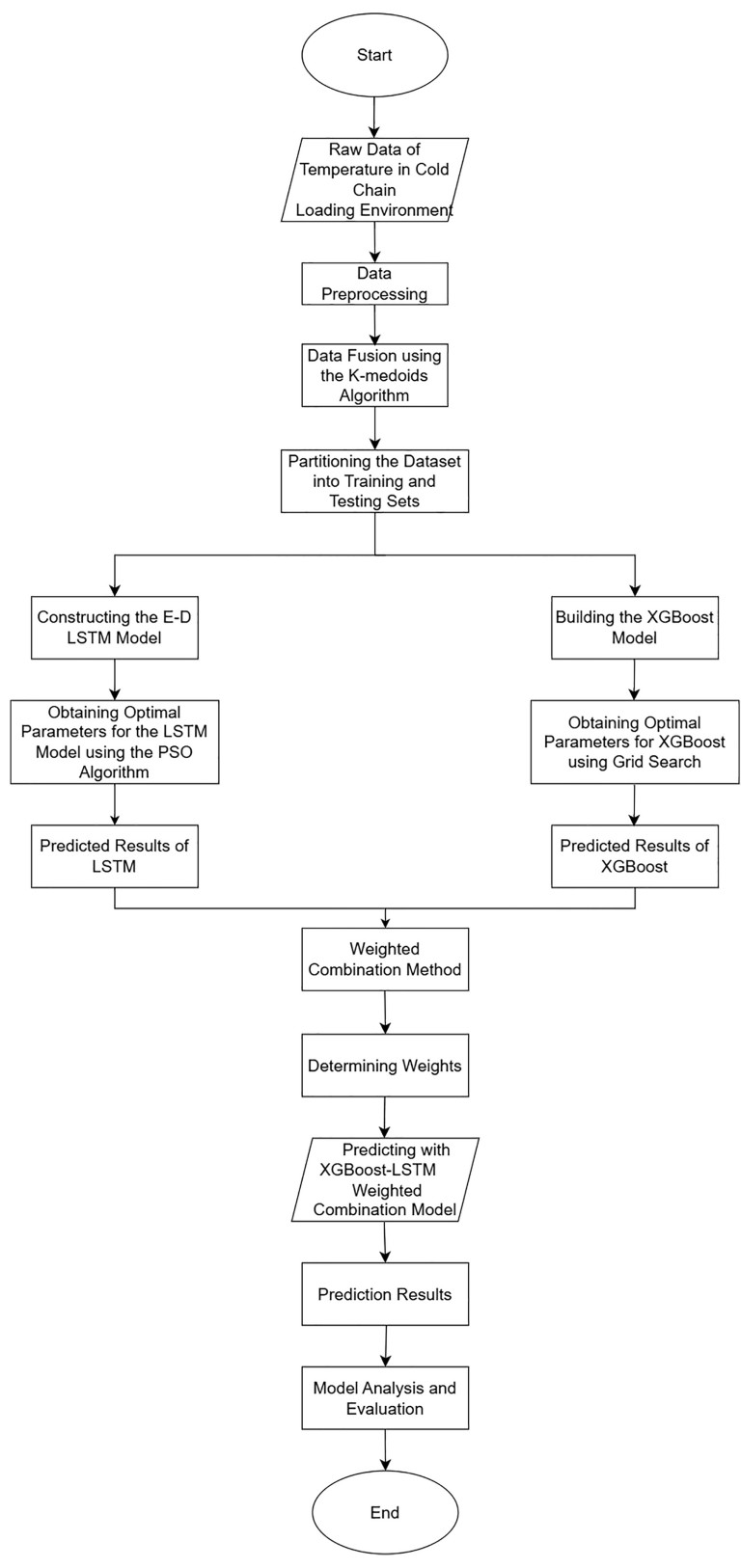

**Figure 5 Flowchart of cold chain loading environment prediction.**

The values $\alpha_1$ and $\alpha_2$ represent the prediction errors of the LSTM and XGBoost models, respectively. The model with smaller error and higher accuracy will have a relatively higher proportion in the ensemble model, whereas a model with larger error will have a lower proportion.

The specific steps are as follows:

1) Collect temperature data from sensors in the remote cold chain vehicle.
2) Preprocess the data, including cleaning the data, removing outliers, filling missing data, and normalizing the data.
3) Apply the K-medoids algorithm to fuse the data and improve its accuracy.
4) Divide the fused data into training and testing sets in proportion.
5) Initialize the LSTM neural network model and optimize its parameters using the PSO algorithm. Then, input the testing set data into the trained LSTM model for prediction, obtaining the predicted data.
6) Initialize the XGBoost model and import the training set data. Perform grid search to obtain the optimal parameters for the XGBoost model. Then, input the testing set data into the trained XGBoost model for prediction, obtaining the predicted data.
7) Combine the two sets of models using the weighted fusion method described above to generate the final prediction results.
8) Introduce evaluation metrics for model assessment and conduct validation and analysis of the prediction results for the two individual models and the combined model. Additionally, to enhance the persuasiveness of the results, this study will compare, validate, and analyze them using models such as random forest and gate recurrent unit (GRU).
9) Figure 5 illustrates the flowchart depicting the prediction process of the cold chain loading environment.

# EXPERIMENTAL ANALYSIS AND DISCUSSION

## Experimental preparation

### Data source

The data utilized in this study was acquired from Zhongkai University of Agriculture and Engineering. Different types of products require distinct storage conditions. For instance, frozen transport is necessary for goods such as pork, beef, lamb, and vaccines, which must be kept at temperatures ranging from −18 °C to −22 °C. In contrast, refrigerated transport is suitable for perishable goods like fruits, vegetables, and fresh milk, where temperatures between 0 °C and 7 °C are optimal. Additionally, temperature-controlled transport, typically maintaining a range of 18 °C to 22 °C, is used for products such as chocolate, pharmaceuticals, and chemical goods. This study focuses on frozen transport as the experimental background, aiming to adjust the cargo compartment's temperature precisely to −18 °C to −22 °C before loading. By doing so, cold chain companies can reduce the operational time of refrigeration systems, save energy, and ensure that products are stored in optimal conditions, minimizing the risk of damage. Given that cooling rates vary with

**Table 1 Temperature data subset.**

| Index (One per Minute) | Temperature 1: Current Room Temperature +23 | Temperature 2: Current Room Temperature +23 | Temperature 3: Current Room Temperature +23 | Temperature 3: Current Room Temperature +23 |
|---|---|---|---|---|
| 1 | −18.5 | −18.8 | −18.5 | −18.6 |
| 2 | −18.3 | −18.3 | −18.1 | −18.2 |
| 3 | −18.5 | −18.5 | −18.2 | −18.5 |
| 4 | −18.2 | −18.3 | −17.9 | −18.1 |
| 5 | −18.1 | −18.0 | −17.9 | −18.2 |

temperature, multiple simulations were conducted to ensure the model's applicability across different thermal environments. Specifically, data were collected under three ambient conditions: 23 °C, 26 °C, and 28 °C. The graph illustrates each cycle, from trough to peak and back to trough, representing one complete cold chain loading simulation. The initial trough corresponds to the rise in temperature after the air conditioning system is switched off while waiting for the cargo to be loaded. The peak reflects the temperature drop once the refrigeration system is activated. Finally, the last trough represents the point at which the cargo loading temperature is achieved. In this experiment, the cargo compartment temperature fluctuated between −20 °C and 30 °C, with the data exhibiting consistent and predictable patterns. A refrigerated truck was employed on the university campus to simulate the entire process of cold chain loading, and the necessary data for this study were collected in real-time. Within the refrigerated truck, data collection was configured to occur at a frequency of one measurement per minute, resulting in a total of nine data sets being collected. Due to space constraints, only a portion of the collected data is presented in the following Table 1.

### Data processing

Upon completion of data collection, the initial step prior to commencing the formal experiment involves preprocessing the raw data. The data preprocessing phase comprises two essential steps: data cleansing and data normalization. Data cleansing entails scrutinizing the raw input data to identify any missing or anomalous values. In the presence of such values, mean imputation is employed to substitute them with the average value of the corresponding column. Mean imputation refers to the process of filling in missing or abnormal data with the mean value. Subsequently, the data undergoes normalization, a procedure that significantly contributes to the accuracy of the models employed. The normalization process is mathematically represented by the following Eq. (18):

$$x'_i = \frac{x_i - \bar{x}}{max(x) - \min(x)}. \tag{18}$$

Finally, the normalized time series dataset is restructured into a supervised learning format. Specifically, the model uses data from the preceding 30 time steps as input features

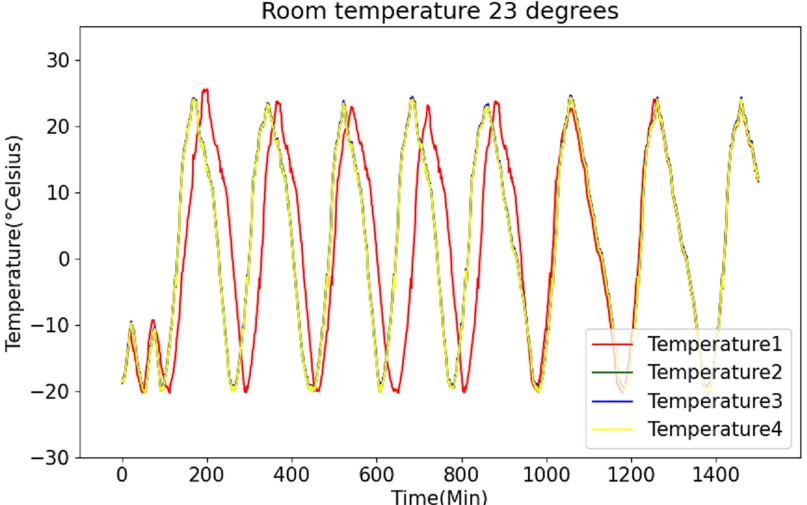

**Figure 6** Raw data curve of room temperature at 23 °C.

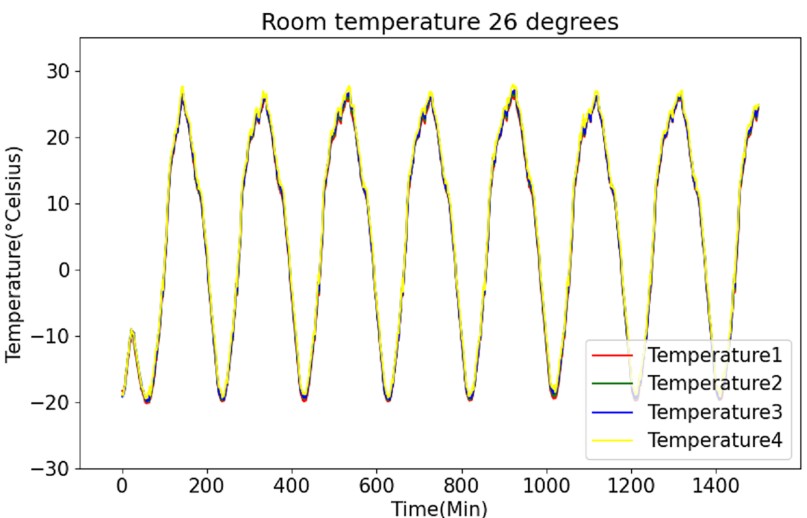

**Figure 7** Raw data curve of room temperature at 26 °C.

to predict the next 15 time steps. This entails leveraging data from the past 30 min to forecast the subsequent 15 min.

## K-medoids data fusion results

Figures 6 and 7 present the original data for ambient temperatures of 23 °C and 26 °C, respectively. These figures indicate periodic temperature variations with a period exceeding 190 to 200 min.

Unlike the K-means algorithm, the K-medoids algorithm does not require the computation of means as centroids. Instead, it selects existing points as medoids during each iteration. This characteristic enables the K-medoids algorithm to perform computations rapidly and efficiently. By employing this method for data fusion on the collected temperature time series, it becomes possible to effectively extract the desired

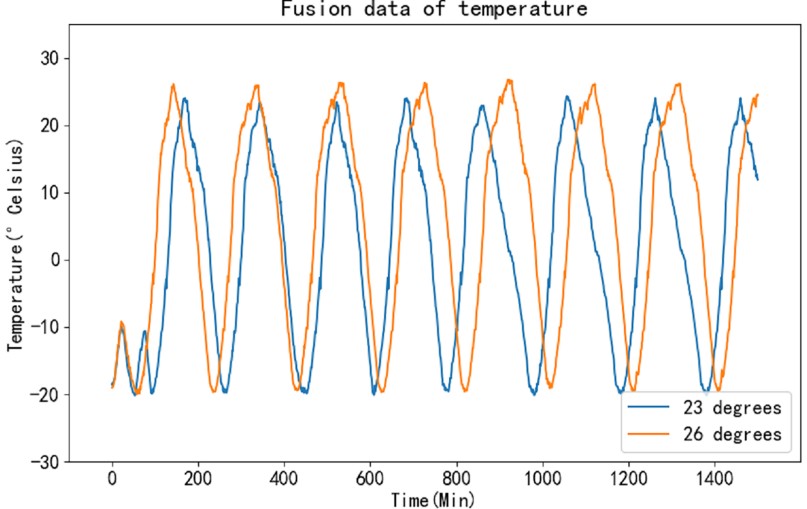

**Figure 8 Fusion curve of raw data.**

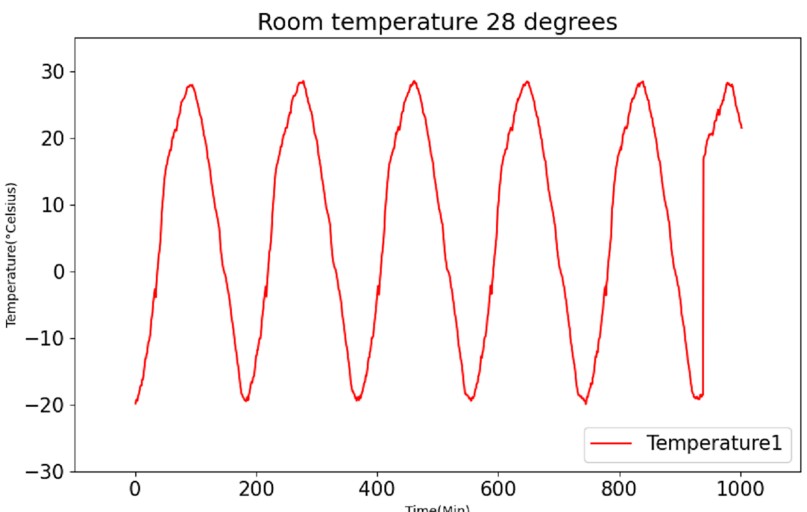

**Figure 9 Raw data curve of room temperature at 28 ℃.**

feature information, thereby enhancing the accuracy of the prediction results. Figure 8 illustrates the fusion results of the original data.

## LSTM neural network prediction results

After employing the K-medoids algorithm for data fusion, the fused data is subsequently subjected to normalization and divided into distinct training and testing sets, following a predetermined ratio. In this experiment, to showcase the universality of the model, the conventional approach to data set partitioning is eschewed. Instead, the training set encompasses data recorded at temperatures of 23 °C and 26 °C, while the testing set comprises data recorded at 28 °C. Figure 9 presents the original data recorded at 28 °C.

Based on Fig. 10, the hidden layer predominantly comprises a recurrent neural network (RNN) constructed using LSTM and Dropout. The intermediate RepeatVector layer serves

# Hidden Layer

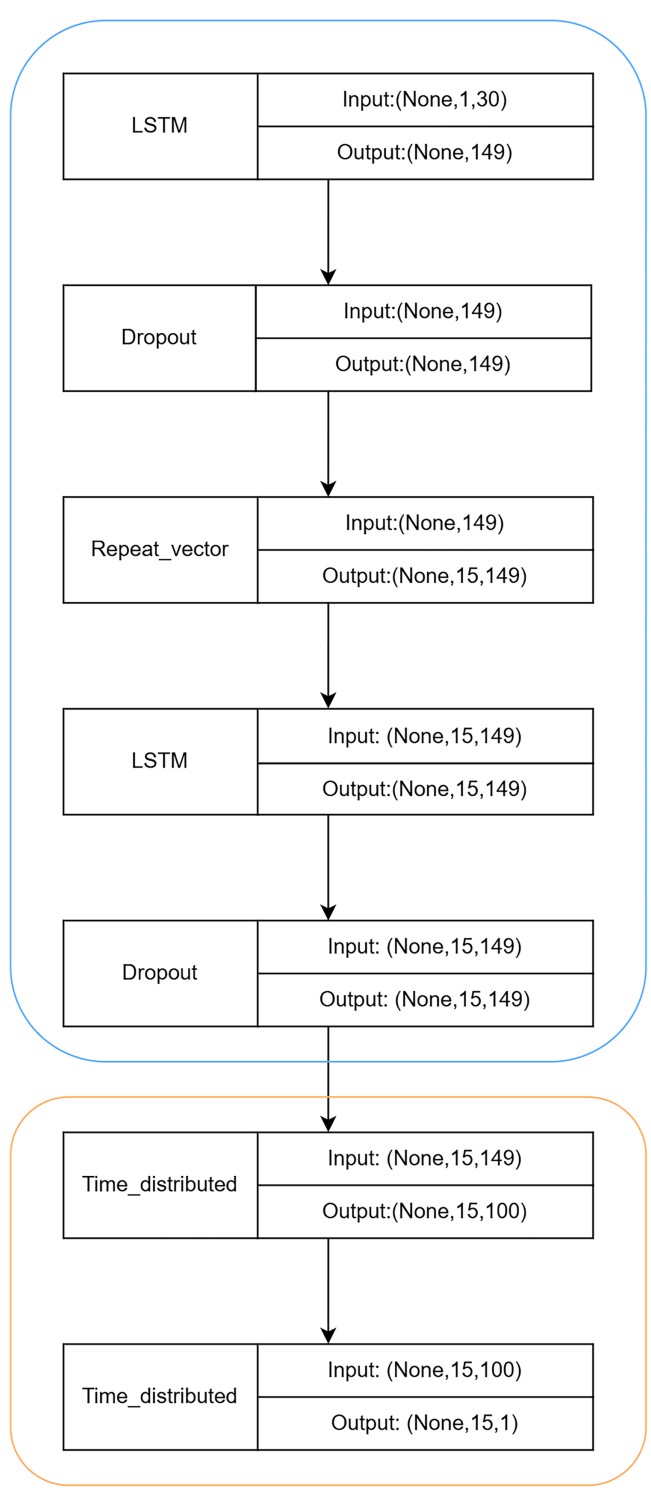

# Output layer

**Figure 10  Hierarchical structure of LSTM.**     

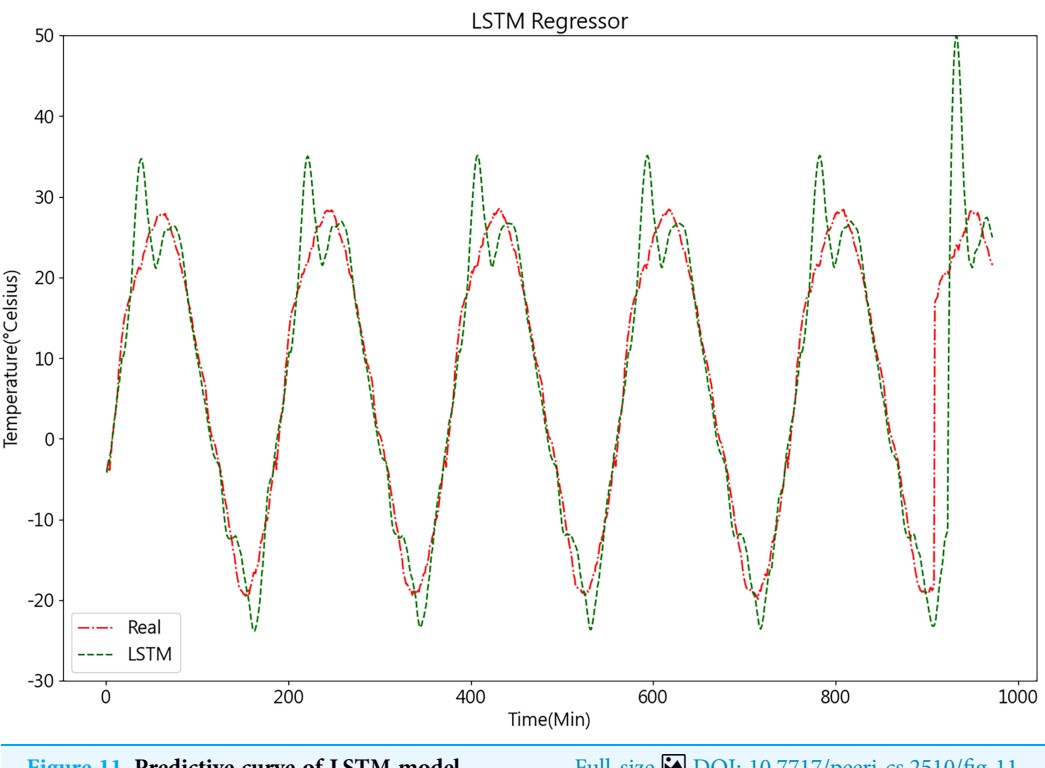

**Figure 11 Predictive curve of LSTM model.**

to establish a connection between the encoder and decoder components. To mitigate convergence challenges associated with deep neural networks, the hidden layer employs a modest configuration of only two layers. The LSTM layer performs computations and predictions, whereas the Dropout layer safeguards against overfitting, thereby enhancing the LSTM model's generalization capabilities.

The output layer will utilize the TimeDistributed layer to process the outputs from the LSTM hidden layer, reducing dimensionality and predicting the final outcome.

Once the LSTM model is constructed, the PSO algorithm is employed to optimize its parameters, aiming to enhance the model's accuracy. The results of the LSTM model are depicted in Fig. 11.

## XGBoost model predictions

Before running the XGBoost model, it is crucial to determine three types of parameters: general parameters, booster parameters, and task parameters. General parameters control the overall functions and are commonly used for tree models or linear models. Booster parameters can control the booster type (tree/regression) at each step. Once the general parameters are set, the corresponding booster parameter types are determined. Booster parameters can influence the model's effectiveness and computational cost. In essence, tuning the XGBoost model largely involves adjusting the booster parameters. Task parameters control the performance of the training objective, such as classification or regression tasks, including binary or multi-class classification.

**Table 2 XGBoost model parameters.**

| Parameter name | Setting result |
| --- | --- |
| max_depth (Tree's maximum depth) | 23 |
| subsample (Random sampling) | 0.3 |
| learning_rate (Learning rate) | 0.12 |
| n_estimators (Maximum number of iterations) | 600 |
| random_state (Specify random seed) | 42 |

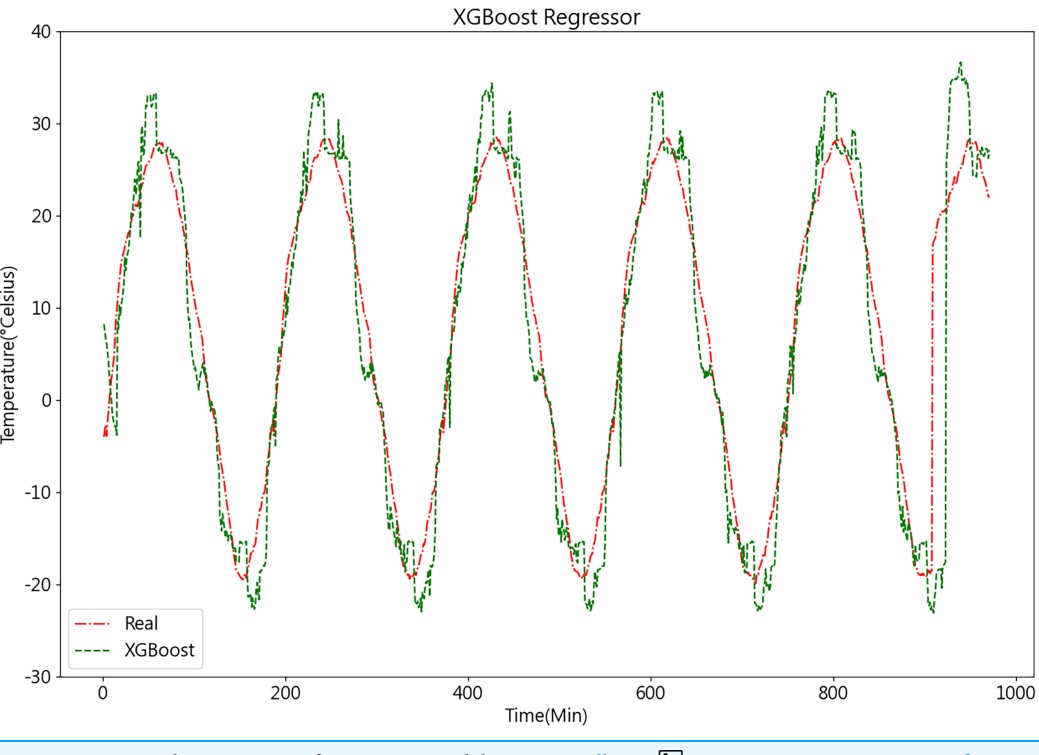

**Figure 12 Predictive curve of XGBoost model.**

These three sets of parameters significantly impact the algorithm's performance. After determining the general and task parameters, the next step involves parameter tuning, specifically the booster parameters. Common booster parameters include learning rate, minimum loss reduction for splitting (gamma), and maximum depth of trees. As the prediction results are directly influenced by the maximum depth of trees, optimizing this parameter is the first step in model tuning. The optimization process begins by assigning initial values to the parameters other than the maximum depth of trees, typically using common or default values. Through multiple iterations and adjustments, the optimal combination of parameters is obtained.

Once the maximum depth of trees is determined, a traversal search method is used to find the best combination of tree depth with other parameters. After the traversal search, the optimal parameter combination obtained is presented in Table 2.

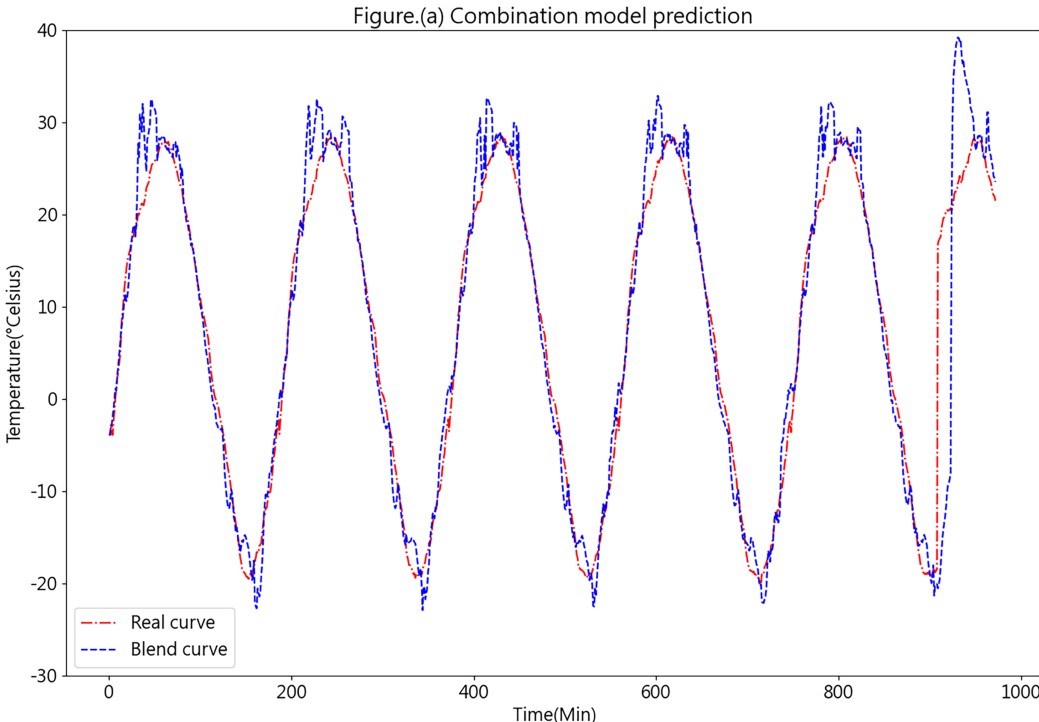

**Figure 13 Predictive curve of the combined K-medoids-LSTM-XGBoost model.**

After acquiring the optimal parameters, the dataset is inputted into the XGBoost model for prediction. The temperature prediction values and actual values derived from the XGBoost model demonstrate their respective trends, as depicted in Fig. 12.

## Prediction results of the combined K-medoids-LSTM-XGBoost model

By employing the variable weight combination method described above, the LSTM model undergoes fine-tuning using the PSO algorithm, while the XGBoost model is tuned through an exhaustive search approach. Following this, the two sets of models are combined to form an ensemble prediction model. Finally, an analysis and prediction are conducted under identical prediction conditions.

Figure 13 illustrates the prediction results obtained from the K-medoids-LSTM-XGBoost ensemble forecasting model.

Figure 13 showcases the prediction performance of the K-medoids-LSTM-XGBoost ensemble forecasting model, revealing a favorable overall performance. While certain prediction errors are evident in specific aspects, such as peaks and valleys, these inconsistencies could be attributed to the limited dataset quantity and the relatively large prediction interval. Such factors may lead to insufficient training. In practical applications, as more data is continuously collected from each farm's cold chain loading, the predictive accuracy of the model is expected to improve significantly.

## COMPARISON AND ANALYSIS

### Comparison and analysis

Model evaluation metrics provide a intuitive way to assess the accuracy of prediction results and serve as a means to evaluate the quality of predictions. Therefore, in order to evaluate the accuracy of the model's predictions, this study will employ three metrics to evaluate the predicted results $y(i)$ against the true values of the dataset.

R-squared.

$$R^2 = 1 - \frac{\sum_{i=1}^{n} (\hat{y}_i - y_i)^2}{\sum_{i=1}^{n} (y_i - \bar{y})^2} \tag{19}$$

Mean absolute error (MAE).

$$MAE = \frac{1}{n} \sum_{i=1}^{n} |\hat{y}_i - y_i| \tag{20}$$

Root mean square error (RMSE).

$$RMSE = \sqrt{\frac{1}{n} \sum_{i=1}^{n} (\hat{y}_i - y_i)^2} \tag{21}$$

### Comparison of predictive results for baseline models

In this experiment, the raw data underwent data fusion using the K-medoids algorithm. The resulting processed data was then utilized as input for comparative models in the prediction process. To facilitate meaningful comparisons, a variant of the RNN known as the GRU algorithm was employed. The GRU algorithm shares many similarities with RNN and provides a highly comparable approach. Additionally, random forests, an ensemble learning algorithm based on decision trees, was chosen as the second comparative model. This choice allows for a reasonable basis of comparison. The prediction results obtained from the GRU algorithm and Random Forests are displayed in the figure below.

From Figs. 11, 12, 14A, and 14B, it can be observed that the prediction outcomes of the GRU and Random Forest models exhibit a similar overall trend as the true curve. However, they lack accuracy in capturing the finer details compared to the LSTM and XGBoost algorithms. Specifically, discrepancies are evident in identifying peaks and valleys. Therefore, in the context of agricultural cold chain loading, the LSTM and XGBoost algorithms demonstrate relative advantages in single-model prediction. Consequently, this study selects the LSTM and XGBoost models as part of the ensemble model.

### Comparison of predictive results for ensemble models

The predicted results of the K-medoids-GRU-Random Forest ensemble prediction model are shown in Fig. 15A.

Figures 15A and 15B clearly demonstrate that the K-medoids-LSTM-XGBoost ensemble prediction model significantly outperforms the K-medoids-GRU-Random

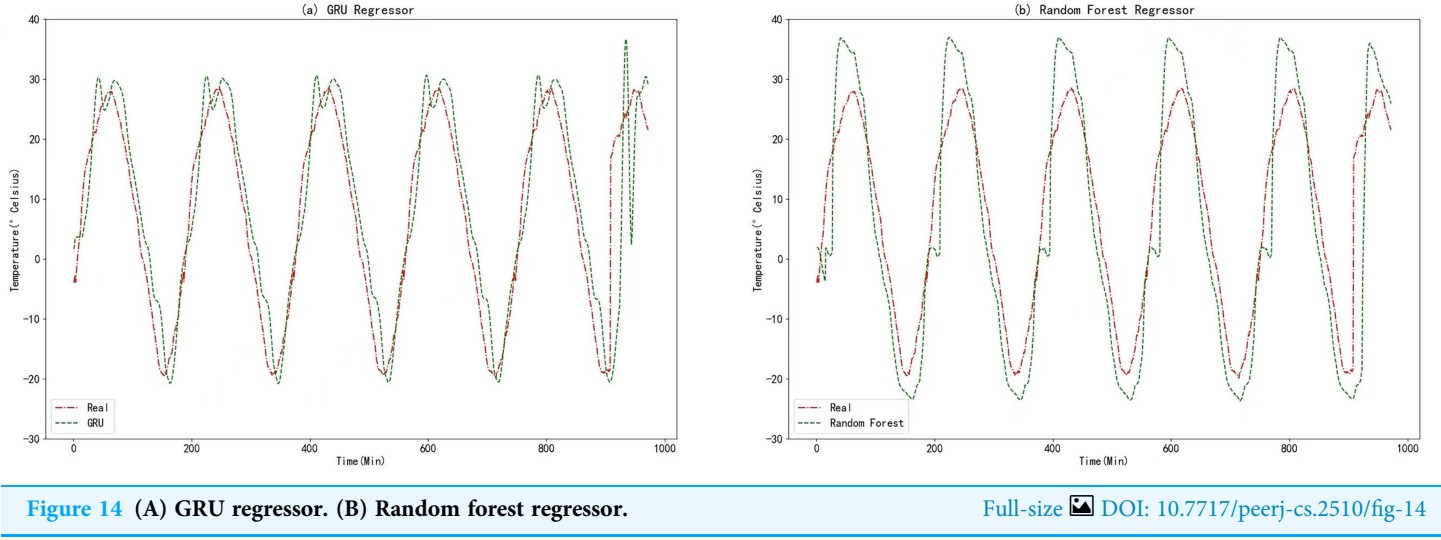

**Figure 14** (A) GRU regressor. (B) Random forest regressor.

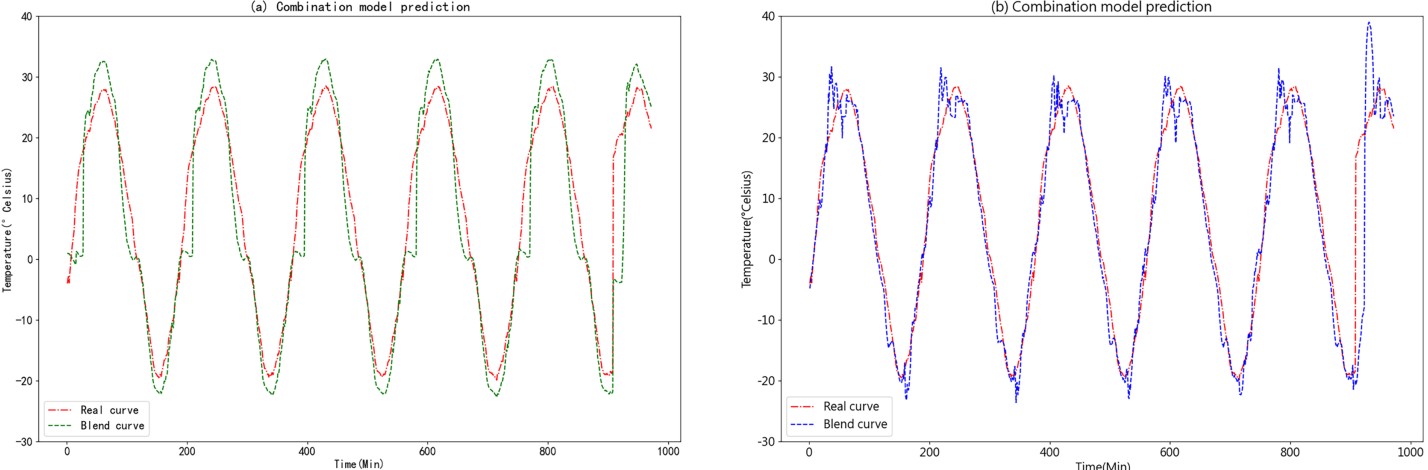

**Figure 15** (A) Predictive curve of the combined K-medoids-GRU-Random Forest model. (B) Predictive curve of the combined K-medoids-LSTM-XGBoost model.

Forest ensemble prediction model in terms of accuracy. This superiority is particularly evident in the fitting of the slopes on both sides of the peaks and valleys, where the K-medoids-LSTM-XGBoost ensemble prediction model exhibits a high degree of overlap. This finding strongly supports the superiority of the K-medoids-LSTM-XGBoost ensemble prediction model in the context of agricultural cold chain loading.

## Practical applications

Based on the comparison between the predicted figures and the model error analysis presented in Table 3, it is evident that the prediction models proposed in this study successfully achieve temperature forecasting within the cold chain loading environment to varying degrees. Upon evaluating the predictive performance of these models, most of

**Table 3 Error in model comparison.**

| Model | MAE | RMAE | R-square |
|---|---|---|---|
| LSTM | 3.7421 | 6.1846 | 0.8539 |
| XGBoost | 3.9952 | 6.4403 | 0.8416 |
| K-medoids-LSTM-XGBoost | 2.5343 | 5.1906 | 0.8971 |
| GRU | 4.4415 | 6.7998 | 0.8234 |
| Random Forest | 6.9001 | 8.8602 | 0.7002 |
| K-medoids-GRU-Random Forest | 4.2694 | 5.9432 | 0.8651 |

them exhibit a substantial overlap between the predicted values and the true value curve in their respective prediction figures. Notably, each combination model consistently outperforms the individual models. Among the ensemble models, the K-medoids-LSTM-XGBoost ensemble prediction model consistently outperforms the K-medoids-GRU-Random Forest ensemble prediction model in terms of MAE, RMAE, and R-square. This study also incorporates the K-medoids-Attention-BiGRU model for comparison. The model integrates an Attention mechanism and bidirectional gated recurrent units (BiGRU). The inclusion of the Attention mechanism allows the model to dynamically assign weights based on the importance of the input data, thereby enabling more accurate identification of key information within the sequence. BiGRU, with its superior ability to capture both forward and backward dependencies in time series, demonstrates improved performance compared to LSTM networks. As a result, this model is particularly well-suited for long-term time series predictions that require capturing both global and local significance, as well as handling complex data dependencies. However, as shown in Table 3, while the K-medoids-Attention-BiGRU model performs effectively, the K-medoids-LSTM-XGBoost model is more suitable for predicting the temperature environment in agricultural cold chain logistics. The experimental results demonstrate that the K-medoids-LSTM-XGBoost prediction model for temperature in the agricultural cold chain loading environment achieves an RMSE, MAE, and R-square of 2.5343, 5.1906, and 0.8971, respectively, indicating its remarkable accuracy in reflecting the actual cold chain loading environment data.

Based on the predictive results, this experiment preemptively controls the temperature environment within cold chain vehicles in practical applications. Through comparative analysis, it is concluded that the algorithm and methods proposed in this article can effectively predict temperature fluctuations inside cold chain vehicles after activating the refrigeration system under various ambient temperatures during agricultural product loading. This allows for an optimal determination of when to initiate the refrigeration system, thereby minimizing the adverse effects of excessively high temperatures on transported goods during loading, reducing energy consumption, and lowering transportation costs. In practical applications, temperature control and real-time monitoring are the core tasks of the entire experiment. To prevent uncontrollable incidents, users can establish a redundancy system to ensure that temperature is

continuously monitored by sensors, even in the event of partial equipment failure. To ensure the system's reliability, regular maintenance, upgrades, and component replacement are necessary. During data transmission, issues such as network congestion or equipment performance problems may lead to transmission delays or even data loss, ultimately affecting the system's real-time prediction and cargo safety. Therefore, optimizing the data transmission protocol by reducing packet size and increasing transmission speed can help mitigate the risk of transmission failures.

## SUMMARY

In summary, the experiments conducted to predict temperature in the cold chain loading environment reveal that both the LSTM model and the XGBoost model outperform the GRU model and the Random Forest model in terms of predictive performance. Additionally, the ensemble prediction model based on K-medoids-LSTM-XGBoost demonstrates superior predictive capabilities compared to the individual prediction models. Moreover, the accuracy of the K-medoids-LSTM-XGBoost ensemble prediction model surpasses that of the similar K-medoids-GRU-Random Forest ensemble prediction model. Therefore, the K-medoids-LSTM-XGBoost ensemble prediction model holds applicability and advantages in the field of agricultural cold chain loading, thereby contributing to the preservation of agricultural products during transportation.

## CONCLUSIONS

In response to the practical situation of cold chain loading, this article proposes a novel prediction method that improves the accuracy of prediction results. Experimental results demonstrate that, in the context of cold chain loading, the model error metrics MAE, MSE, and R-squared of the combined prediction model based on K-medoids-LSTM-XGBoost are 2.5343, 5.1906, and 0.8971, respectively. These data reflect the model's ability to accurately predict temperature trends for a future time period in the cold chain loading environment. Consequently, managers can flexibly adjust the timing of activating the refrigeration mode based on the predicted data. This approach not only reduces the risk of agricultural product spoilage but also minimizes energy consumption in the cold chain process, promoting energy efficiency and environmental sustainability. However, due to limited data, it is not possible to accurately predict temperature data for various temperature ranges within a future time period. Additionally, an insufficient number of temperature sensors, uneven sensor distribution, inadequate precision, or equipment aging can result in inaccurate detection of the internal temperature of the vehicle compartment. Frequent opening and closing of the doors during loading in high-temperature conditions further affects the accuracy of temperature predictions. These factors—limited data, sensor inefficiencies, and external environmental influences—contribute to reduced accuracy and validity of predictive models. To mitigate these limitations and enhance the reliability and precision of temperature forecasts, this study aims to continuously optimize the temperature control system and explore new predictive algorithm models. Future research will also focus on predicting temperature variations in dynamic environments, such as during loading and unloading operations or under extreme weather conditions (*e.g.*, severe

cold or heat). Nevertheless, with continued application in real-world scenarios and the accumulation of data under different temperature conditions, it is believed that the accuracy of the prediction results can be significantly improved.

### Funding
This work was supported by the Guangdong Provincial Young Innovative Talents Program in Ordinary Universities (No. 2017KQNCX097) and the Guangzhou Municipal General Scientific Research Program (No. 201904010233), the Research and Innovation Project for Graduate Education Reform at Zhongkai University of Agriculture and Engineering (No. KA220160228) and the Guangdong Rural Science and Technology Special Envoy Project (No. KTP20240633). The funders had no role in study design, data collection and analysis, decision to publish, or preparation of the manuscript.

### Grant Disclosures
The following grant information was disclosed by the authors:
Guangdong Provincial Young Innovative Talents Program: 2017KQNCX097.
Guangzhou Municipal General Scientific Research Program: 201904010233.
Research and Innovation Project for Graduate Education Reform: KA220160228.
Guangdong Rural Science and Technology Special Envoy Project: KTP20240633.

### Competing Interests
The authors declare that they have no competing interests.

### Author Contributions
- Zhijie Luo conceived and designed the experiments, performed the experiments, analyzed the data, prepared figures and/or tables, authored or reviewed drafts of the article, and approved the final draft.
- Wenjing Liu conceived and designed the experiments, performed the experiments, analyzed the data, performed the computation work, authored or reviewed drafts of the article, and approved the final draft.
- Jianhao Wu performed the experiments, analyzed the data, prepared figures and/or tables, and approved the final draft.
- Huang Aiqing performed the experiments, analyzed the data, authored or reviewed drafts of the article, and approved the final draft.
- Jianjun Guo conceived and designed the experiments, performed the experiments, analyzed the data, authored or reviewed drafts of the article, and approved the final draft.

### Data Availability
The raw data is available in the Supplemental File.

## Supplemental Information

Supplemental information for this article can be found online at http://dx.doi.org/10.7717/peerj-cs.2510#supplemental-information.

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
