# Peer review of "Prediction of cold chain loading environment for agricultural products based on K-medoids-LSTM-XGBoost ensemble model"

_PeerJ Computer Science, doi:10.7717/peerj-cs.2510_

## Round 0.1 · original submission · Major Revisions

Please add the requested details regarding the experimental configurations and computational resources.

·

Basic reporting

The article demonstrates generally clear and professional English throughout, with appropriate technical language used for the subject matter. It provides sufficient background and context on cold chain logistics and temperature prediction models, referencing relevant prior literature. The structure follows a standard scientific format with clear sections, and includes relevant figures and tables to illustrate the methodology and results. Raw data is mentioned as being collected and processed, though it's not explicitly stated if it was shared. The study presents a self-contained piece of research, with results directly addressing the proposed prediction model's performance. While the article focuses on an applied machine learning approach rather than formal mathematical proofs, it does provide clear explanations of the algorithms and models used. Overall, the article meets most of the basic reporting criteria for a scientific publication in this field, though some minor improvements in clarity and data sharing practices could be beneficial.

Experimental design

This article presents original primary research on predicting cold chain loading environments for agricultural products using a novel ensemble model combining K-medoids, LSTM, and XGBoost algorithms. The research question is well-defined, addressing the need for accurate temperature prediction in cold chain logistics to improve quality preservation and reduce energy consumption. The study clearly identifies the knowledge gap in existing prediction models and proposes a new approach to fill this gap. The investigation appears to have been conducted rigorously, employing established machine learning techniques and following ethical standards in data collection and analysis. The methodology is described in considerable detail, including data preprocessing, model architecture, parameter optimization, and evaluation metrics, which should allow for replication by other researchers. However, some additional information on the specific cold chain vehicle setup and data collection process might be beneficial for complete reproducibility. Overall, the study meets the criteria for original, relevant, and technically sound research with a well-defined methodology.

Validity of the findings

The study presents a novel approach to predicting cold chain loading environments using a combined K-medoids-LSTM-XGBoost model, which appears to be an original contribution to the field. While the impact is not explicitly assessed, the authors provide a clear rationale for their research and demonstrate its potential benefit to cold chain logistics. The underlying data seems to have been collected through a simulated refrigerated truck experiment, with the authors mentioning data preprocessing and fusion techniques. However, it's not explicitly stated whether all raw data has been made available in a repository, which could be a point for improvement. The statistical analysis appears sound, with multiple evaluation metrics (MAE, RMSE, R-squared) used to assess model performance. The conclusions are well-stated and directly linked to the original research question of improving temperature prediction accuracy in cold chain loading environments. The authors appropriately limit their claims to the supporting results, comparing their ensemble model's performance to individual models and similar ensemble models. They also acknowledge limitations due to the limited dataset and suggest potential for further improvement with more data collection in real-world scenarios, demonstrating a balanced and scientifically sound approach to their findings.

Additional comments

While the article presents a novel approach to predicting cold chain loading environments with promising results, there are several areas where improvements could be made. The authors should explicitly state whether all raw data has been made available in a public repository, as this is crucial for scientific reproducibility. Additionally, more detailed information about the cold chain vehicle setup and data collection process would enhance the study's replicability.

The discussion of limitations could be expanded beyond the acknowledgment of the limited dataset. A more comprehensive analysis of potential limitations and their implications would strengthen the paper's credibility. Furthermore, while the simulation-based approach is valuable, the authors could discuss plans or challenges for applying their model to real-world cold chain logistics scenarios, enhancing the study's practical relevance.

The comparative analysis could be broadened to include more state-of-the-art methods in the field, providing a more comprehensive evaluation of the proposed model's performance. Although not directly applicable to this type of study, a brief discussion on potential ethical implications in the real-world application of their model could be valuable. Lastly, expanding on future research directions based on their findings would help situate their work in the broader context of ongoing research in this field. Addressing these points could further strengthen the overall quality and impact of the article.

Reviewer 2 ·

Basic reporting

I like the concept of this research, but their are gaps that need to be clarified before this can be published.
I have a few problems with the work. First, I can't quite determine what system is being modelled and why. Specifially, there is a lack of a detailed description of the experimental setup and data collection process. The temperatures that were recorded, what was measured? Why do they cycle between -20C and 30C, these are extreme ranges. Why would a truck do this. Also the temperature cycles are very cyclic and consistent. What were the input parameters that were being used to make forecasts?

What was the the exact time frame for predictions (e.g., how far into the future the model can predict).

Please also details on the computational resources required to run the model

Include a discussion of potential challenges in implementing this system in real-world cold chain logistics operations

Experimental design

Very few details were given regarding the experimental design. The authors need to detail what the input parameters are, what the predictions actally represent.

Validity of the findings

It does seem like the findings are valid. However, again, the authors do not provide sufficient information to understand what is being predicted. This makes the value of findings somewhat unclear.

---

## Round 0.2 · accepted · Accept

The reviewers, and I, agree on the fact you revised the paper according to their suggestions

·

Basic reporting

The paper can be accepted in current form

Experimental design

The paper can be accepted in current form

Validity of the findings

The paper can be accepted in current form

Additional comments

The paper can be accepted in current form

Reviewer 2 ·

Basic reporting

The paper is sufficent for publication.

Experimental design

The paper is sufficent for publication.

Validity of the findings

The paper is sufficent for publication.